# Community Actions and Insights in the Battle against COVID-19 at the Start of the Pandemic: A District Study Observation from Medan, Indonesia

**DOI:** 10.3390/ijerph21040444

**Published:** 2024-04-04

**Authors:** Nadya Keumala Fitri, Meliani Meliani, Kartini Marpaung, Raden Andika Dwi Cahyadi, Ranti Permatasari, Cut Meliza Zainumi, Inke Nadia Diniyanti Lubis

**Affiliations:** 1Faculty of Medicine, Universitas Sumatera Utara, Medan 20155, Indonesia; nadyafitri520@gmail.com (N.K.F.); radendwicahyadi@gmail.com (R.A.D.C.); 2Department of Clinical Pathology, Faculty of Medicine, Universitas Sumatera Utara, Medan 20155, Indonesia; melmeliani321@gmail.com (M.M.); rantirizal@yahoo.com (R.P.); 3Department of Paediatrics, Faculty of Medicine, Universitas Sumatera Utara, Medan 20155, Indonesia; kartinimarpaung08@gmail.com; 4Department of Anaesthesiology and Intensive Care, Faculty of Medicine, Universitas Sumatera Utara, Medan 20155, Indonesia; cut.meliza@usu.ac.id

**Keywords:** attitudes, COVID-19, educational interventions, Indonesia, knowledge level, North Sumatra, pandemic response, prevention practices, public perception, social media

## Abstract

Background: As of 17 June 2020, the WHO confirmed 8,061,550 COVID-19 cases globally, with Indonesia reporting 40,400 cases and North Sumatra over 932 cases. The rising infection rates have led to increased deaths, highlighting the urgency for public understanding of virus transmission. Despite information dissemination efforts, North Sumatra has not seen a reduction in cases, emphasizing the need for a unified approach to combat the pandemic. Objective: This study aims to investigate the relationship between public perception and practices regarding COVID-19 prevention in Medan, North Sumatra. Methods: A cross-sectional study will be conducted using a combined questionnaire from two previous studies conducted at the start of the pandemic. Results: Among 200 participants, social media was the favored source for prevention information. Participants exhibited above-average knowledge (67.5%) but predominantly below-average attitudes toward prevention (64.5%). However, most residents practiced correct prevention measures (75.5%). Conclusions: Despite possessing adequate knowledge, negative attitudes toward prevention suggest a need for educational interventions to address misconceptions and promote positive behaviors. Such interventions could enhance the community’s response to COVID-19 transmission during the pandemic.

## 1. Introduction

Pandemics have far-reaching impacts on various socioeconomic factors and quality of life. Global poverty, for instance, has increased for the first time in a generation. Survey data show that income losses were particularly pronounced among youth, women, the self-employed, and casual workers with lower levels of formal education. Similarly, businesses, especially smaller and informal ones with limited access to formal credit, experienced severe income losses due to the pandemic [1]. This situation necessitates ongoing attention not only from the government but also from scientists and healthcare professionals who address each case as it emerges.

Moreover, pandemics have had a detrimental effect on the health-related quality of life of the general population. Several factors, including age, sex, marital status, education, chronic diseases, confinement, and financial constraints, have been identified as influencing the quality of life [2]. During the first wave of the pandemic, prevalent symptoms such as insomnia, anxiety, and depression were majorly reported in multiple countries, with the addition of studies expressing some incidence of PTSD [3]. Therefore, it is crucial for the general population to play a role in mitigating the transmission of the virus while protecting themselves. This includes ensuring that the public has a consistent and thorough understanding of the disease, from awareness to preventive practices, to effectively implement preventive measures.

Shifting focus to COVID-19, this respiratory infectious disease manifests pneumonia-like symptoms, including fever, dry cough, and shortness of breath [4]. Transmission transpires through droplets during coughing or sneezing and via contact transmission from the surfaces of inanimate objects [5]. Identified in Wuhan, Hubei Province, China in December 2019, Indonesia confirmed two cases on 2 March 2020 [6]. Escalating to over 40,000 cases within four months, Indonesia has become the third country in Southeast Asia with the highest number of COVID-19 cases. Globally, the total number of cases has surpassed 6 million [6,7]. The mortality rate of COVID-19 in Indonesia displayed significant diversity, influenced by factors like varying incidence rates, different levels of pre-existing health conditions, the capacity of the healthcare system to respond to the pandemic, and the socioeconomic makeup of the population. The majority of COVID-19 cases and deaths were concentrated on Java Island, which is more developed and home to 152 million people, making up 56% of Indonesia’s total population [8]. Recent studies conducted in Jakarta, the capital city of Indonesia, indicated that the virus had a disproportionate impact on older individuals with existing chronic health issues. Additionally, areas within Jakarta that had lower rates of vaccination, higher levels of poverty, and higher population densities experienced more severe effects from the virus [9]. In response to this surge, the Indonesian government implemented large-scale social restrictions and empowered local authorities to enforce measures to curb further transmission in their regions [10].

To implement proper prevention practices, one’s knowledge and attitude should be solidified to enact these practices. Several factors influence these practices, including knowledge, attitude, occupation, education level, and place of residence. A study conducted in 30 provinces in China demonstrated exemplary results, with an average accurate response rate of 91.2% for knowledge, 98% for attitude, and 96.8% for practices [11]. Similarly, studies in both the USA and the UK showed that participants generally had good knowledge of the main modes of disease transmission and common symptoms [12]. In Indonesia, specifically in the capital city of Jakarta, the results were adequate, with average scores of 83% for knowledge, 70.7% for attitude, and 70.3% for practices [13].

In Indonesia, efforts to heighten disease awareness among the community involved the use of social media to disseminate information on preventive measures [14]. These preventive actions aim to disrupt the transmission chain, safeguarding individuals and the community. Measures include mask usage, regular handwashing, social distancing, self-isolation, and temperature screening at public places [14]. Although large-scale preventive measures, such as the nationwide lockdown, contributed to a shorter duration of lockdown and a lower increase in the case growth rate in the post-lockdown era, it also resulted in harsh restrictions on the economy and people’s lives. The people adopted other preventive measures instead, such as mass mask-wearing, patient/suspected case isolation strategies, physical distancing, and contact tracing [15,16].

Despite these efforts, the number of cases persists, and the public’s attitude towards the pandemic does not align with the recommended practices. This study seeks to assess the knowledge, attitude, and practices of the community in Medan City concerning the prevention of COVID-19 transmission. 

## 2. Materials and Methods

### 2.1. Study Design

This is a cross-sectional survey conducted in the Medan Selayang subdistrict, Medan City, North Sumatra province, Indonesia on 9 May 2020. The study site was the first subdistrict to report COVID-19 cases in the city. The participants were given half a kilogram of sugar as a token of appreciation for completing the survey.

### 2.2. Recruitment Procedure 

The sub-district selected for this questionnaire was determined based on the progression of COVID-19 transmission. We opted for the Medan Selayang subdistrict, as it was among the first to report a COVID-19 case and has the highest number of inpatients and persons under surveillance from that area [17]. Consequently, the government has designated this sub-district as a red zone. It has been chosen as the target population for our study.

The minimum sample size of this study was calculated with the formula for a finite population size [18], using a ±7% margin of error and a confidence level of 95% from a population of 106,150 people, resulting in a minimum sample size of 196 people. A printed questionnaire was given to the visitors during the weekly market at the Sub-District Head Office. Health protocols, including the usage of surgical masks and face shields and the implementation of physical distancing, were performed during the survey. A post to wash the hands and alcohol-containing hand sanitizers were also provided. The following day, a leaflet containing COVID-19 information was distributed to the community, in hopes of educating the community regarding COVID-19 transmission prevention. Informed consents in Bahasa Indonesia were obtained from all participants.

### 2.3. Study Instrument 

The questionnaire comprised 32 questions assessing knowledge, attitudes, and practices related to COVID-19 prevention. These questions were combined and adapted from earlier studies, titled ‘Knowledge and Perceptions of COVID-19 Among the General Public’ and ‘Perceptions of the adult US population regarding the novel coronavirus outbreak’ [12,19]. The combined questionnaire is then directly translated into Bahasa Indonesia with modifications to suit the social nuances of the community in the district. Prior to the commencement of the study, the questionnaire underwent validity testing, which yielded a Cronbach’s alpha score of 0.74. Following the validity test, it was determined that 10 questions were valid for assessing knowledge, 9 questions for attitudes, and 13 questions for practice. The Likert scale measurement will be categorized based on the scores, with scoring as 0 = strongly disagree/disagree/neutral and 1 = agree/strongly agree. The categorical results will be dichotomized based on the mean cutoff. Knowledge, attitude, and practice will use the mean cutoff of 9, 6, and 13, respectively.

### 2.4. Statistical Analysis

Data were analyzed using the SPSS program (version 24; SPSS Inc., Chicago, IL, USA). In order to assess the differences in knowledge, attitudes, and practices related to sociodemographic characteristics, logistic regression analysis was used, with a *p*-value < 0.05 as a statistically significant parameter. 

## 3. Results

### 3.1. Demographic Characteristic

A total of 200 residents of the Medan Selayang sub-district were enrolled in the study. Baseline characteristics are presented in Table 1. 

The average age of participants was 40 years old, with the age range of 11–86 years old, with a majority of women (61%), having the highest education level of high school (59%) and were in the other category (45%). Most of the participants (77.5%) used social media as the main source to receive COVID-19 information, with the least coming from healthcare workers (27%) (Table 2).

### 3.2. Knowledge on COVID-19

A majority of participants had an above-average (67.5%) knowledge about COVID-19 transmission prevention and its clinical features (Table 3).

The average score was 8.95 (SD = 1.63) from a total of 13 questions. The most correctly answered question was the knowledge of the existence of COVID-19 (99%), and early signs of COVID-19 (96%, Table 3). Most participants recognized cough (44.4%) and fever (42.1%) as early signs of COVID-19 but not shortness of breath (11.4%). Despite the basic awareness of COVID-19 symptoms, there was a common misconception that people with other health problems are less likely to have worsened conditions (19.5%). 

### 3.3. Attitude towards COVID-19

The majority of the community (64.5%) had a low perceived risk score for COVID-19, with a mean of 5.475 out of 9 (SD = 2.01). The most controversial opinion was that one is not more susceptible than the other (76%) and that smoking (63%) does not increase the risk of infection (Table 4).

### 3.4. Practice Assessment

In contrast to the result of the attitude, this study’s findings present an above-average practice of COVID-19 (75.5%, Table 5).

However, a significant number of participants were found to engage in behaviors not recommended for preventing COVID-19 transmission, as outlined in Table 6. This includes activities like regular exercise (45%), consuming garlic regularly (39%), and regularly taking antibiotics (34%) (Table 6).

### 3.5. Statistical Analyses

The study employed logistic regression analysis to test the relationship between variables, with statistical significance defined as a *p*-value < 0.05 (Table 7). In the context of attitude, for individuals aged 60 or older, there is a significant positive association with attitude, with a coefficient of 1.298 and a *p*-value of 0.042. The odds ratio of 3.662 indicates that the odds of having a positive attitude are higher for this age group compared to the reference group. Within this age range, 55.6% of the population scored above average, with an average score of six.

Additionally, there is a positive association in practice with people aged 60 and above, as indicated by a coefficient of 1.662. The *p*-value of 0.011 suggests that this association is statistically significant. Furthermore, the odds ratio of 5.268 indicates that the likelihood of having a positive practice is higher in this age group compared to the reference group. Within this age group, 72.2% scored above average, with the average score of the group scoring 13. This analysis revealed that media usage amongst other variables in the categories, was not statistically significant.

## 4. Discussion

During the survey, conducted six weeks after the first reported case in Medan city, COVID-19 was a relatively recent and emerging concern in Indonesia, primarily concentrated in Java, even though the first case in North Sumatra was reported on 18 March 2020, only 16 days apart from the first reported case in Jakarta (2 March 2020) [4,14]. By 9 May 2020, only 132 cases had been identified, with 12 deaths among 2.9 million citizens, and the sub-district in Medan City that has the most positive patients is Medan Selayang sub-district, with a total of >20 patients being treated [14]. This research will compare the relationship between the levels of perception and behavior of the people of North Sumatra Province in preventing the transmission of COVID-19. It aims to provide an overview of the community’s response to the pandemic and ways to address it at the grassroots level.

Our study focused on evaluating the community’s knowledge, attitudes, and practices in one of the first sub-districts to report COVID-19 cases in Medan. The participants, predominantly aged between 30 and 39 years, with a high school education or higher, and mostly in occupations such as housewives and drivers, indicated social media as the primary source for preventive measures and COVID information. A quantitative analysis study based on the Protection Motivation Theory (PMT) showed that perceived risk, e-health literacy, public awareness, and health experts’ participation influence public protective behavior when using social media to share COVID-19-relevant content. This result can offer guidance for advocating health practices and information to the public [20]. Despite social media’s role in providing information and serving as a means to connect during the pandemic, it has also been linked to increased anxiety and potentially negative effects on protective behaviors and self-efficacy [21,22,23].

The widespread use of social media poses a challenge to ensuring the accuracy of information. Participants also exhibited a common misconception regarding the susceptibility of individuals with other health conditions to COVID-19, indicating a lack of awareness about the factors that increase the risk of contracting the virus. In another study conducted in Indonesia, COVID-19 questionnaires were distributed using the social media app WhatsApp. The results revealed a stark contrast to our study, with the majority of participants demonstrating a high level of knowledge (98%) and positive attitudes (96%). One notable difference between the two studies is the educational background of the participants. In our study, the majority have a high school education (59%), whereas in the other study, the majority have a bachelor’s degree/S1 (57.7%) [24]. Similarly, a study conducted in Jakarta in mid-June 2020 also showed positive attitude results, with over 70% of participants exhibiting good attitudes and 66% having a higher educational background [13]. These studies, conducted through online platforms, consistently show positive outcomes among participants with higher educational levels, including diplomas and undergraduate degrees.

In contrast, our current study distributes physical questionnaires, resulting in a participant pool that is more diverse in terms of educational and socioeconomic backgrounds. This diversity reflects a broader representation of grassroots communities, rather than those who are more educationally privileged. These differing characteristics between studies may explain the varying abilities of participants to discern information circulating about the pandemic. These findings highlight a misunderstanding among participants regarding their vulnerability to COVID-19. Participants’ perceptions of their likelihood of infection differ from other studies that show a high perceived risk, even though it is in line with the result of study instrument reference number 5 [19]. This discrepancy could be due to participants’ unfamiliarity with pandemic settings, leading them to perceive COVID-19 as a mild illness similar to common respiratory infections. 

Another study in a different sub-district of Medan showed a higher percentage of positive attitudes (95%), despite having a similar majority of participants with senior high school education. However, the occupation demographics differed, with 30% of the participants in the other study being business owners, whereas this study was predominantly composed of participants in other categories, including housewives and online taxi drivers [25].

The majority (75%) believes that COVID-19 could significantly damage their health; they simultaneously perceive themselves as at low risk of infection, indicating a misunderstanding regarding vulnerability to the virus. This indicates a prevalent negative perception among participants regarding the risk of COVID-19 infection. Another study explored the perceived risk as a whole in Indonesia, resulting in the same low-risk perception [26]. 

Additionally, a substantial portion (63%) disagreed with the idea that smoking increases the risk of COVID-19 infection. Many participants do not agree that factors such as smoking history increase their risk of infection. This contrasts with research suggesting an increase in smoking tendencies since the pandemic began [27]. Furthermore, the fact is that smokers are more likely to experience adverse effects from a COVID-19 infection than non-smokers, with one study displaying an increased risk of 1.53 times [28].

Even if so, participants demonstrated good knowledge, with the majority (60%) answering correctly. Knowledge of symptoms, method of transmission, and what the term close contacts means are the questions with the highest number of correct answers. Acknowledging the symptoms and method of transmission is essential for preventing the disease by assisting in early detection; the results are similar to studies conducted in other countries [29,30,31]. 

Regular handwashing with soap or using alcohol-based hand sanitizer, minimizing in-person contact and practicing social distancing, wearing a face mask in public places, and avoiding going to public places unless essential are among the precautions, even if it is only 62% of the participants’ answers contains those activities, which is lower than the questionnaire reference [10]. On the other hand, less than 20% of the participants know that people with comorbidities (ex: diabetes and hypertension) are more susceptible. A study has proven how COVID-19 infection impairs glycaemic control in diabetes by increasing inflammation and altering the nature of the immune system’s response and, therefore, increases the risk of complications in diabetic patients, leading to the development of thromboembolism or cardiovascular and respiratory failure [32]. Hypertension alongside other cerebrovascular diseases is independently associated with in-hospital mortality and intensive care unit (ICU) admission, although, given the established role of hypertension as the principal risk factor for cardiovascular diseases, this association may not necessarily be limited to COVID-19. In addition, a large fluctuation in BP in COVID-19 patients with poor prognosis may simply reflect their critical conditions [33,34,35].

Addressing these misconceptions and attitudes is crucial in developing strategies to prevent the transmission of COVID-19. Public health interventions should aim to educate the public about the risks associated with COVID-19 and promote behaviors that reduce transmission. The study identified several behaviors among participants that are not recommended for preventing the transmission of COVID-19, as shown in Table 6. These include regular exercise (45%), regular consumption of garlic (39%), and regular use of antibiotics (34%). These findings align with a previous study conducted in England, which reported similar rates of antibiotic use (36%) to this study (34%) [12]. This suggests widespread misinformation regarding effective prevention measures for COVID-19.

Both attitude and practice showed a correlation with age, specifically where the age is above 60 years old. This result is similar to a previous study reporting older people have adopted prevention practices for COVID-19 more than the other age group [31], albeit 50% of the age group is above average in knowledge and 61% answered correctly regarding whether having comorbidities can affect susceptibility. This is higher than other age groups. The elderly are in a riskier position, with aging bodies and commonly carrying comorbidities; navigating through the pandemic requires them to be more aware and vigilant towards possible transmission [36]. 

The study’s limitations include its cross-sectional method, which limits its ability to show changes over time. It focused on only one of the districts with high case numbers, potentially limiting its generalizability. Having been conducted at the beginning of the pandemic may represent the fresh panic wave of the pandemic. But, it is susceptible to changes as time goes on within the pandemic, and the government advances to prevent transmission. Future research should consider the dynamic for a more detailed understanding of informed policy revisions and their implications.

## 5. Conclusions

In conclusion, our study in one of Medan’s early COVID-19-affected sub-districts reveals a concerning lack of awareness among participants regarding the virus’s risk factors. Despite good overall knowledge, there were notable gaps, particularly regarding the increased susceptibility of individuals with comorbidities. These findings underscore the need for targeted public health interventions to educate the community about COVID-19 risks and promote effective preventive behaviors, especially among older individuals at higher risk due to comorbidities.

## Figures and Tables

**Table 1 ijerph-21-00444-t001:** Sociodemographic characteristics.

Characteristics		Sample
		Total (*n* = 200)
		*n* (%)
Age (years)		40.24 ± 13.73
Age classification		
	20–29 years old	36 (18)
	30–39 years old	63 (31.5)
	40–49 years old	49 (24.5)
	50–59 years old	34 (17)
	>60 years old	18 (9)
Gender		
	Woman	122 (61.0)
	Man	78 (39.0)
Work		
	Private employees	11 (5.5)
	Civil servants (PNS)	15 (7.5)
	Business owners	84 (42.0)
	Other	90 (45.0)
Education		
	S1	14 (7.0)
	Senior high school	118 (59.0)
	Junior high school	42 (21.0)
	Elementary school	26 (13.0)

**Table 2 ijerph-21-00444-t002:** Source of information in understanding the COVID-19 outbreak.

Resources	*n* (%)
Social media	155 (77.5)
Broadcast media	116 (58.0)
Friends/neighbors/relatives	58 (29.0)
Health workforce	54 (27.0)

**Table 3 ijerph-21-00444-t003:** Knowledge results.

Question	Answer	*n* (%)
Are you aware of the COVID-19 outbreak?	Yes	198 (99.0)
In your opinion, which is the most appropriate definition of the COVID-19 virus?	COVID-19 is a respiratory disease and is a contagious disease	144 (72.0)
In your opinion, what are the early signs of COVID-19?	Cough, fever, and shortness of breath	192 (96.0)
	Contains at least one of the following: nosebleeds, difficulty defecating, frequent urination	8 (4.0)
How COVID-19 is transmitted	Through someone’s coughs and sneezes	141 (70.5)
The correct statement regarding ‘close contact’ with COVID-19	Direct contact with someone’s respiratory fluids	158 (79.0)
Which age groups are more likely to contract COVID-19?	Mature	154 (77.0)
Will someone with diseases such as high blood pressure and diabetes be more easily infected with COVID-19?	Yes	39 (19.5)
Correct statement regarding COVID-19 treatment management	There is no specific treatment that cures it, nor a vaccine	126 (63.0)
What methods are effective in preventing transmission of COVID-19 to yourself and others	Contains one of the following: wearing a mask, washing hands, using disinfectant, staying home if sick, covering mouth if coughing, avoiding touching eyes, mouth, and nose with unwashed hands, avoiding direct contact with someone who is sick	124 (62.0)
	Contains one of the following: getting flu and pneumonia vaccines, taking antibiotics, exercising regularly, eating onions regularly, avoiding eating meat, being careful when opening packages or letters	76 (38.0)
Appropriate measures are taken by the government to prevent the spread of COVID-19	All of the following: quarantine all people arriving from abroad for 14 days, cancel all domestic and international flights, measure the body temperature of all people from door to door, close all schools, prohibit all mass gatherings, such as sports competitions, concerts, birthday events, requiring people to wear masks when outside the home, requiring all people to stay at home except when seeking medical care or looking for food	40 (20)
Mean ± SD		8.95 ± 1.63
Above Average		135 (67.5)
Below Average		65 (33.5)

**Table 4 ijerph-21-00444-t004:** Attitudes results.

No.	Attitude	Answer	
		Agree * (%)	Disagree * (%)
1	My health will suffer serious damage if I contract COVID-19	149 (74.5)	51 (25.5)
2 **	If I get another disease, I will not go to the hospital to avoid getting COVID-19 in the hospital	80 (40)	120 (60)
3	COVID-19 will have a devastating impact on my environment	165 (82.5)	35 (17.5)
4	COVID-19 will spread to all corners of Indonesia	112 (56)	88 (44)
5	I am more likely to get COVID-19 than other people	48 (24)	152 (76)
6	I can protect myself from COVID-19 more than anyone else	137 (68.5)	63 (31.5)
7	Going out and being in large crowds can increase the spread of COVID-19	165 (82.5)	35 (17.5)
8 **	Having a history of smoking does not increase the risk factor for developing COVID-19	74 (37)	126 (63)
9	Touching the mouth and nose of the face can be a place of entry for COVID-19 infection	165 (82.5)	35 (17.5)
Mean ± SD		5.47 ± 2.01	
Above Average		71 (35.5)	
Below Average		129 (64.5)	

(*): The Likert scale will be classified based on scores, with a score of 0 = strongly disagree/disagree/neutral; 1 = agree/strongly agree. (**): Modified the statement by removing the word “don’t” in the table to display consistent values for “agree” and “disagree.

**Table 5 ijerph-21-00444-t005:** Recommended behavior in preventing transmission of COVID-19.

No.	Behavior	Answer *	
		Yes (%)	No (%)
Recommended behavior			
1	Using a mask	197 (98.5)	3 (1.5)
2	Washing hands	198 (99.0)	2 (1.0)
3	Using disinfectant	186 (93.0)	14 (7.0)
4	Stay home if you are sick	165 (82.5)	35 (17.5)
5	Cover your mouth if you cough/sneeze	195 (97.5)	5 (2.5)
6	Avoid touching your eyes, nose, and mouth with unwashed hands	170 (85.0)	30 (15.0)
7	Avoid direct contact with sick people	117 (58.5)	83 (41.5)
Mean ± SD		13.14 ± 0.92	
Above Average		151 (75.5)	
Below Average		49 (24.5)	

(*): This column shows how the response options were grouped to summarize the variable categories into a dichotomous measure, with Regularly and Sometimes summarized in Yes while Rarely and Never in No.

**Table 6 ijerph-21-00444-t006:** Unnecessary behavior in preventing transmission of COVID-19.

No.	Behavior	Answer *	
		Yes (%)	No (%)
Unnecessary behavior			
1	Get a flu/pneumonia vaccine	59 (29.5)	141 (70.5)
2	Taking antibiotics	68 (34.0)	132 (66.0)
3	Exercise regularly	90 (45.0)	110 (55.0)
4	Eat onions regularly	78 (39.0)	122 (61.0)
5	Avoid consuming meat	38 (19.0)	162 (81.0)
6	Be careful when opening the delivery/postal package	62 (31.0)	138 (69.0)

(*): This column shows how the response options were grouped to summarize the variable categories into a dichotomous measure, with Regularly and Sometimes summarized in Yes while Rarely and Never in No.

**Table 7 ijerph-21-00444-t007:** Statistical Analyses.

Attitude							
Variable		Coef (B)	S.E	*p*	Odds Ratio (Exp (B))	95% C.I. for EXP(B)	
						Lower	Upper
Age	≥60	1.298	0.639	0.042	3.662	1.047	12.806
**Practice**							
Variable		Coef (B)	S.E	*p*	Odds Ratio (Exp (B))	95% C.I. for EXP(B)	
						Lower	Upper
Age	≥60	1.662	0.652	0.011	5.268	1.466	18.925

## Data Availability

The data presented in this study are available upon request from the corresponding author. The data are not publicly available due to privacy.

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
