# Peer review of "Community Actions and Insights in the Battle against COVID-19 at the Start of the Pandemic: A District Study Observation from Medan, Indonesia"

_ijerph, 2024, doi:10.3390/ijerph21040444_

Round 1
Reviewer 1 Report
Comments and Suggestions for Authors
This paper provides an adequate accounting of what people in that Sumatra district know about COVID-19. However, the paper would be much better with one or more of the following:
1. A better literature review that indicates what people in other Indonesian districts know about COVID, or in other countries. The study needs a better context so the results can be compared to something. Right now we don't know if these results are better or worse than expected.
2. A better study would have compared the results in this district with results from another district that differed in some meaningful ways, e.g., demographic differences or cultural differences, or disease impact differences. I would recommend collecting data in another carefully selected district so we can understand the impact of social media use in preventing the disease going forward. As the study stands we only have data from one district and it's difficult to determine what these data mean.
Author Response
Thanks to the reviewer for assessing this manuscript. Here are my replies for the revised version attached on this note (Please see the attachment):
1. A better literature review that indicates what people in other Indonesian districts know about COVID, or in other countries. The study needs a better context so the results can be compared to something. Right now we don't know if these results are better or worse than expected.
I have added the contextual materials in the Introduction part of the manuscript within the lines of (73-81)
2. A better study would have compared the results in this district with results from another district that differed in some meaningful ways, e.g., demographic differences or cultural differences, or disease impact differences. I would recommend collecting data in another carefully selected district so we can understand the impact of social media use in preventing the disease going forward. As the study stands we only have data from one district and it's difficult to determine what these data mean.
Although i am unable to obtain the data from another sub district within the exact same time frame of the first wave for this study as a comparison, i have added studies from other sub districts and parts of Indonesia as a form of comparison, within the lines of (245-250,256-273,279-284)
Thank you very much for your revision and i hope this suffices, if there is anything else, please let me know.
Best Regards.

Reviewer 2 Report
Comments and Suggestions for Authors
Thanks to the authors for sharing their manuscript. Please see my comments:
1. I would recommend specifying the instrument assessing knowledge, attitudes, and practices related to COVID-19 prevention in the abstract.
2. The introduction is very short and does not contain any references to the huge number of studies devoted to the negative consequences of the COVID-19 pandemic. I recommend expanding the introduction (at least to include several papers with meta-analyses and systematic reviews).
3. I don't see any ethical considerations. Have the authors received permission from the ethics commission to conduct this study? If this is not possible, then it is worth pointing out that the study was conducted in compliance with the principles of the Helsinki Declaration.
4. The authors write that they used a measure that evaluates knowledge, attitudes, and practices related to COVID-19 prevention and refer to sources. The name of this instrument remains unclear. The authors also write that they translated this instrument into Bahasa Indonesia, but do not give information about the translation (it would be direct or reverse, etc.).
5. There are no limitations and prospects for research in the manuscript, it is considered a mandatory component of an empirical article.
I think the research is interesting, but the manuscript needs major revision.
Author Response
Thanks to the reviewer for assessing this manuscript. Here are my replies for the revised version attached on this note (Please see the attachment):
- I would recommend specifying the instrument assessing knowledge, attitudes, and practices related to COVID-19 prevention in the abstract.
- I have added the specific description of the questionnaire with the lines of 'A cross-sectional study will be conducted using a combined questionnaire from two previous studies conducted at the start of the pandemic' within the methods of the Abstract in the lines 19-21
- The introduction is very short and does not contain any references to the huge number of studies devoted to the negative consequences of the COVID-19 pandemic. I recommend expanding the introduction (at least to include several papers with meta-analyses and systematic reviews).
- I have added several references in the introduction to explain the severity of the COVID-19 impact on the general public in indonesia, within the lines of 32-53, 60-70, and 86-91
- I don't see any ethical considerations. Have the authors received permission from the ethics commission to conduct this study? If this is not possible, then it is worth pointing out that the study was conducted in compliance with the principles of the Helsinki Declaration.
- I have added the Institutional Review Board Statement and the Informed Consent Statement within the lines 360-364
- The authors write that they used a measure that evaluates knowledge, attitudes, and practices related to COVID-19 prevention and refer to sources. The name of this instrument remains unclear. The authors also write that they translated this instrument into Bahasa Indonesia, but do not give information about the translation (it would be direct or reverse, etc.).
- I have added the description of the questionnaire used in the study and how it was directly translated with modifications, within the Materials and Methods. The questionnaire is available upon request as a supplementary file. Lines (120-125 and 350-351)
- There are no limitations and prospects for research in the manuscript, it is considered a mandatory component of an empirical article.
- I have added it within the Discussion, within the lines of (335-341)
Thank you very much for your revision and i hope this suffices, if there is anything else, please let me know.
Best Regards.

Round 2
Reviewer 1 Report
Comments and Suggestions for Authors
This study is much improved since it provides more background information and a richer discussion section. The data are presented clearly and the results are appropriately compared to other similar studies which places the results in context. As a result of these changes, the manuscript is now ready for publication.
Reviewer 2 Report
Comments and Suggestions for Authors
Thanks to the authors for having finalized the manuscript in accordance with all the comments. I recommend the manuscript for publication.